# Female sex is linked to a stronger association between sTREM2 and CSF p-tau in Alzheimer's disease

Davina Biel [1,✉], Marc Suárez-Calvet [2,3,4,5], Anna Dewenter[1], Anna Steward [1], Sebastian N Roemer [1,6], Amir Dehsarvi [1], Zeyu Zhu[1], Julia Pescoller[1], Lukas Frontzkowski[1,7], Annika Kreuzer[7], Christian Haass [8,9,10], Michael Schöll [11], Matthias Brendel[7,8,9,12] & Nicolai Franzmeier [1,9,11,12]

## Abstract

In Alzheimer's disease (AD), Aβ triggers p-tau secretion, which drives tau aggregation. Therefore, it is critical to characterize modulators of Aβ-related p-tau increases which may alter AD trajectories. Here, we assessed whether factors known to alter tau levels in AD modulate the association between fibrillar Aβ and secreted $p$-$tau_{181}$ determined in the cerebrospinal fluid (CSF). To assess potentially modulating effects of female sex, younger age, and ApoE4, we included 322 ADNI participants with cross-sectional/longitudinal $p$-$tau_{181}$. To determine effects of microglial activation on $p$-$tau_{181}$, we included 454 subjects with cross-sectional CSF sTREM2. Running ANCOVAs for nominal and linear regressions for metric variables, we found that women had higher Aβ-related $p$-$tau_{181}$ levels. Higher sTREM2 was associated with elevated $p$-$tau_{181}$, with stronger associations in women. Similarly, ApoE4 was related to higher $p$-$tau_{181}$ levels and faster $p$-$tau_{181}$ increases, with stronger effects in female ApoE4 carriers. Our results show that sex alone modulates the Aβ to p-tau axis, where women show higher Aβ-dependent p-tau secretion, potentially driven by elevated sTREM2-related microglial activation and stronger effects of ApoE4 carriership in women.

Keywords Alzheimer's Disease; Microglia; sTREM2; p-tau; Sex Differences
Subject Category Neuroscience

## Introduction

Beta-amyloid (Aβ) and tau are the hallmark pathologies of Alzheimer's disease (AD), ensuing neurodegeneration and cognitive decline (Jack et al, 2018). Previously it was shown that Aβ-oligomers trigger the neuronal secretion of soluble phosphorylated tau (p-tau) (Jin et al, 2011), which in turn drives the aggregation and trans-synaptic spread of neurofibrillary tau tangle pathology (Pichet Binette et al, 2022). These findings indicate that Aβ-related p-tau increases are critical for the subsequent development of tau aggregates. While the sequence of the amyloid cascade is well-established (Glenner and Wong, 1984; Hardy and Selkoe, 2002; Selkoe and Hardy, 2016), the progression rate of Aβ-initiated tau accumulation differs significantly between patients, causing heterogenous disease trajectories and dynamics across patients (Dujardin et al, 2020; Komarova and Thalhauser, 2011; Landau et al, 2022). The aim of the present study is therefore to determine modulating factors of the Aβ to p-tau axis, which can help identify factors that accelerate or attenuate tau progression and determine targets to attenuate p-tau secretion and subsequent tau aggregation. To this end, we specifically focused on factors that have been previously associated with an increased risk of tau pathology in AD, which are female sex, microglial activation, younger age, and ApoE4, which is the main genetic risk factor for AD.

Accumulating evidence indicates that women are more severely affected by AD than men (Buckley et al, 2019b; Fisher et al, 2018; Laws et al, 2018; Levine et al, 2021; Mosconi et al, 2017; Nebel et al, 2018; Vest and Pike, 2013), accounting for two-thirds of AD dementia cases in the US (Alzheimer's, 2014). Previous work revealed that women show increased levels of CSF total tau and p-tau (Hohman et al, 2018), tau deposition (Buckley et al, 2019b), and a faster tau accumulation rate compared to men (Smith et al, 2020). Recently it was found that faster tau accumulation in women was facilitated by a stronger association between Aβ fibrils and

[1]Institute for Stroke and Dementia Research (ISD), University Hospital, LMU Munich, Munich, Germany. [2]Barcelonaβeta Brain Research Center (BBRC), Pasqual Maragall Foundation, Barcelona, Spain. [3]IMIM (Hospital del Mar Medical Research Institute), Barcelona, Spain. [4]Servei de Neurologia, Hospital del Mar, Barcelona, Spain. [5]Centro de Investigación Biomédica en Red de Fragilidad y Envejecimiento Saludable (CIBERFES), Madrid, Spain. [6]Department of Neurology, University Hospital, LMU Munich, Munich, Germany. [7]Department of Nuclear Medicine, University Hospital, LMU Munich, Munich, Germany. [8]German Center for Neurodegenerative Diseases (DZNE), Munich, Germany. [9]Munich Cluster for Systems Neurology (SyNergy), Munich, Germany. [10]Chair of Metabolic Biochemistry, Biomedical Center (BMC), Faculty of Medicine, LMU Munich, Munich, Germany. [11]University of Gothenburg, The Sahlgrenska Academy, Institute of Neuroscience and Physiology, Department of Psychiatry and Neurochemistry, Mölndal and Gothenburg, Sweden. [12]These authors contributed equally: Matthias Brendel, Nicolai Franzmeier. ✉E-mail: Davina.Biel@med.uni-muenchen.de

soluble p-tau in women compared to men, suggesting early Aβ-dependent tau secretion as a critical turnover point for the observed sex differences in AD (Wang et al, 2024). Although the driving mechanisms behind these findings are not fully clear, differences in sex hormones (Sundermann et al, 2020) and inflammatory processes (Casaletto et al, 2022) are assumed to play an underlying role in the manifestation of sex differences in AD. In this context, analysis of post-mortem brain tissue of older adults show that microglial activation mediated the association between Aβ plaque pathology and neurofibrillary tau tangles in women but not in men (Casaletto et al, 2022). An association between microglial activation and the accumulation and spread of tau in AD has been previously shown in vitro (Brelstaff et al, 2021; Maphis et al, 2015) and in vivo (Pascoal et al, 2021; Vogels et al, 2019), thus microglial activation may play an early key role in the amyloid cascade (Pascoal et al, 2021). From a mechanistic point of view, microglia are the brains innate immune cells, which react to hazardous stimuli with the release of pro-inflammatory cytokines. Once the hazard stimulus has been eliminated, microglia return to their original homeostatic resting state. However, in chronic inflammation, microglia lose their capability to return back to their homeostatic state, resulting in neurotoxicity and tissue damage (Bivona et al, 2023). The Triggering Receptor Expressed on Myeloid Cell 2 (TREM2) has been associated with a shift of microglia from homeostatic to disease associated states (Keren-Shaul et al, 2017; Krasemann et al, 2017), parallels PET-assessed microglial activation (Brendel et al, 2017), and is thus a well-established proxy for microglial activation (Ewers et al, 2020; Ewers et al, 2019; Franzmeier et al, 2020). We and others showed recently that increased CSF sTREM2 levels are associated with higher levels of p-tau in early phases of sporadic AD (Suarez-Calvet et al, 2019), and mediate Aβ-related p-tau increases in earliest Aβ fibrillization (Biel et al, 2023). Thus, microglial activation might have a pivotal role in the early pathogenesis of AD, with different effects in men and women.

A further potential modulator of the Aβ to p-tau axis might relate to the patient's age. In addition to the effects of sex and sTREM2-related microglial activation, studies show that younger age at symptom onset is associated with a worsened prognosis and faster tau accumulation in sporadic AD (Frontzkowski et al, 2022; Koedam et al, 2008; van der Vlies et al, 2009). However, it remains unclear, whether younger age modulates the Aβ to p-tau axis towards a faster p-tau increase, which would support the view of a more aggressive form of AD when patients enter the amyloid cascade at younger age (Koedam et al, 2008; Touroutoglou et al, 2023).

Finally, genetic predispositions might play a critical role for early tau pathology. The ε4 allele of the ApoE-encoding apolipoprotein (ApoE4) is the main genetic risk factor for sporadic AD, and carriership of the ApoE4 risk allele has been linked to an oversupply of cholesterol, ensuing accelerated Aβ production (Lee et al, 2021), neuroinflammation (Ophir et al, 2005), impaired myelination (Blanchard et al, 2022), and tau-mediated neurodegeneration (Shi et al, 2017). Further, ApoE4 carriership is associated with lower levels of testosterone (Hogervorst et al, 2002), a sex hormone which is mostly expressed in males and associated with anti-inflammatory processes (Bianchi, 2019; Ota et al, 2012) and cholesterol clearance (Kilby et al, 2021). Previously, we showed that ApoE4 drives Aβ-related tau

accumulation at lower levels of Aβ pathology, suggesting that in ApoE4 carriers, tau accumulation starts earlier than in ApoE4 non-carriers (Steward et al. 2023). In addition, several studies indicate that female ApoE4 carriers show enhanced levels of soluble CSF total tau (Altmann et al, 2014; Babapour Mofrad et al, 2020; Buckley et al, 2019a; Damoiseaux et al, 2012; Hohman et al, 2018) and p-tau (Babapour Mofrad et al, 2020; Hohman et al, 2018) compared to their male counterparts, hence, the association between ApoE4 carriership and Aβ-dependent tau might be further modulated by sex.

Together, female sex, microglial activation, younger age, and ApoE4-related genetic predisposition for AD have been previously associated with increased tau burden. The main aims of the present study were therefore to test, (i) whether sex, sTREM2-related microglial activation, younger age or ApoE4 modulate the Aβ to p-tau axis, and (ii) whether the effects of sTREM2, younger age or ApoE4 are stronger in women than in men.

# Results

## Sample characteristics

To assess whether sex, age, or ApoE4 modulates the Aβ to p-tau axis, we included 144 CN Aβ− controls, as well as 178 participants across the AD spectrum as defined by Aβ+ status [CN/MCI/Demented=48/115/15] with cross-sectional and longitudinal CSF p-tau$_{181}$ data. The average follow-up time from baseline for p-tau$_{181}$ was $3.54 \pm 1.97$ years. As expected, the Aβ+ group showed higher baseline levels of p-tau$_{181}$, higher p-tau$_{181}$ change rates and had a higher likelihood of ApoE4 positivity. For the assessment of CSF sTREM2 as a potential modulator of the Aβ to p-tau$_{181}$ axis, only cross-sectional data were available. 160 CN Aβ− and 294 Aβ+ [CN/MCI/Demented=41/160/93] participants with sTREM2 and cross-sectional p-tau$_{181}$ data were included. As for the longitudinal sample, the Aβ+ group showed higher p-tau$_{181}$ levels. For detailed descriptive statistics, see Table 1.

## Higher sTREM2 levels and ApoE4 as risk factors for increased p-tau$_{181}$ levels

We tested first whether sex, CSF sTREM2, age, or ApoE4 are associated with increased cross-sectional or longitudinal CSF p-tau$_{181}$ levels. To that end, we calculated main effects using ANCOVAs for nominal potential modulators (i.e., sex, ApoE4) and linear regression for metric potential modulators (i.e., sTREM2, age). We found that higher sTREM2 was related to higher p-tau$_{181}$ levels ($T = 10.098$, $p < 0.001$, $p_{FDR} < 0.001$, partial $R^2 = 0.186$; Fig. 1C), and that ApoE4 was related to higher p-tau$_{181}$ levels ($F = 45.398$, $p < 0.001$, $p_{FDR} < 0.001$, $\eta^2 = 0.13$; Fig. 1F) and faster p-tau$_{181}$ increases ($F = 8.915$, $p = 0.003$, $p_{FDR} = 0.009$, $\eta^2 = 0.03$; Fig. 1G). Neither sex nor age were linked to constitutively higher p-tau$_{181}$ levels (sex: $F = 2.845$, $p = 0.093$, Fig. 1A; age: $T = 1.047$, $p = 0.296$, Fig. 1D) or faster p-tau$_{181}$ increases (sex: $F = 1.611$, $p = 0.205$, Fig. 1B; age: $T = 0.938$, $p = 0.349$, Fig. 1E). The results are summarized in Table 2. In line with previous work, we found that sTREM2-related microglial activation and ApoE4 positivity are related to higher levels in p-tau$_{181}$, suggesting a critical role of both factors in the onset of tau pathology.

**Table 1. Demographics.**

| | Controls (Aβ−) | AD spectrum (Aβ+) | *p*-values |
|---|---|---|---|
| **Modulators sex, age, education, ApoE4** | | | |
| N | 144 | 178 | |
| Clinical status (CN/MCI/Dem) | 144/0/0 | 48/115/15 | <0.001 |
| Sex (male/female) | 71/73 | 85/93 | 0.869 |
| Age in years | 72.72 (6.57) | 73.35 (6.57) | 0.394 |
| Years of education | 16.97 (2.38) | 16.07 (2.57) | 0.001 |
| CSF Aβ$_{1-42}$ (pg/ml) | 1517.16 (552.09) | 760.11 (315.95) | <0.001 |
| CSF p-tau$_{181}$ (pg/ml) | 19.46 (7.22) | 33.94 (15.17) | <0.001 |
| Amyloid-PET (centiloid) | −7.87 (11.80) | 76.73 (35.77) | <0.001 |
| ApoE4 status (non-carrier/carrier) | 119/25 | 66/112 | <0.001 |
| Follow-up CSF p-tau$_{181}$ (mean years) | 3.72 (2.18) | 3.40 (1.77) | 0.145 |
| CSF p-tau$_{181}$ change rate | 0.38 (0.34) | 0.75 (0.62) | <0.001 |
| **Modulator sTREM2** | | | |
| N | 160 | 294 | |
| Clinical status (CN/MCI/Dem) | 160/0/0 | 41/160/93 | <0.001 |
| Sex (male/female) | 79/81 | 165/129 | 0.201 |
| Age in years | 72.60 (6.41) | 73.81 (7.32) | 0.079 |
| Years of education | 16.91 (2.39) | 15.85 (2.71) | <0.001 |
| CSF Aβ$_{1-42}$ (pg/ml) | 1411.73 (396.05) | 631.87 (174.24) | <0.001 |
| CSF p-tau$_{181}$ (pg/ml) | 19.73 (7.46) | 34.52 (15.17) | <0.001 |
| CSF sTREM2 (pg/ml) | 3940.74 (1992.21) | 3966.61 (2210.14) | 0.902 |
| Amyloid-PET (centiloid) | −7.22 (12.08) | 82.55 (33.31) | <0.001 |

Values are presented as mean (SD); *p*-values were derived from ANOVAs for continuous measures and from Chi-squared tests for categorical measures.

Aβ status (−/+) was defined on global amyloid-PET SUVRs.

*CN* cognitively normal, *MCI* mild cognitive impairment, *Dem* demented.

## Female sex is associated with stronger amyloid-dependent p-tau$_{181}$ secretion

Next, we determined whether sex, CSF sTREM2, age, or ApoE4 specifically modulate the Aβ to p-tau axis and increase Aβ-dependent p-tau$_{181}$ secretion, which may translate into faster downstream tau aggregation and spread. Here, we found that solely sex modulates the association between centiloid and p-tau$_{181}$, where women compared to men showed higher Aβ-related p-tau$_{181}$ levels (T = 2.746, *p* = 0.006, p$_{FDR}$ = 0.026; partial R$^2$ = 0.023; Fig. 2A) and faster p-tau$_{181}$ increases (T = 2.169, *p* = 0.031; partial R$^2$ = 0.015; Fig. 2B) which reached trend level significance after FDR correction (p$_{FDR}$ = 0.092). Neither sTREM2, age, or ApoE4 modulated Aβ-related p-tau$_{181}$ levels (sTREM2: T = 0.422, *p* = 0.673, Fig. 2C; age: T = −0.240, *p* = 0.810, Fig. 2D; ApoE4: T = −0.513, *p* = 0.608, Fig. 2F) or faster p-tau$_{181}$ increases (age: T = 0.516, *p* = 0.606, Fig. 2E; ApoE4: T = −1.409, *p* = 0.160, Fig. 2G). The results are summarized in Table 2. The results indicate that sex

is associated with modulation of the Aβ to p-tau axis, suggesting a stronger association between Aβ-dependent p-tau$_{181}$ secretion and eventually faster tau aggregation in women compared to men. In contrast, our analysis did not provide statistical evidence that sTREM2, age, or ApoE4 significantly modulate the Aβ to p-tau axis.

## sTREM2 promotes higher amyloid-dependent p-tau$_{181}$ levels in women compared to men

Next, we aimed to test whether female sex is associated with a higher vulnerability for detrimental effects of potential modulators on CSF p-tau$_{181}$ levels, and specifically, the Aβ to p-tau axis. To that end, we determined interaction effects of each potential modulator (i.e., CSF sTREM2, age, ApoE4) and sex on cross-sectional or longitudinal p-tau$_{181}$ levels. The analysis revealed a significant effect of sex on the association between sTREM2 and p-tau$_{181}$, with women showing a stronger association between sTREM2 and p-tau$_{181}$ levels compared to men (T = 2.177, *p* = 0.030, p$_{FDR}$ = 0.045; partial R$^2$ = 0.011; Fig. 3A). Post hoc analyses calculating separate linear regression analyses for men and women further revealed higher partial R$^2$ values in women (T = 8.286, *p* < 0.001, partial R$^2$ = 0.253) vs. men (T = 5.978, *p* < 0.001, partial R$^2$ = 0.132), supporting a stronger association between sTREM2 and p-tau$_{181}$ in women than in men. In addition, we detected a significant interaction between ApoE4 carriership and sex on p-tau$_{181}$, where female ApoE4 carriers compared to male ApoE4 carriers showed higher levels of p-tau$_{181}$ (T = 2.185, *p* = 0.030, p$_{FDR}$ = 0.045; partial R$^2$ = 0.015; Fig. 3B) but no faster p-tau$_{181}$ increases (T = 1.119, *p* = 0.264). There was no significant effect of sex on the association between age and p-tau$_{181}$ levels (T = −1.005, *p* = 0.315) or p-tau$_{181}$ increases (T = −0.735, *p* = 0.463). The findings show that the associations between sTREM2-related microglial activation as well as ApoE4 status on p-tau$_{181}$ levels are stronger in women than in men.

As a last step, we assessed whether sex further modulates the Aβ to p-tau axis by interacting with potential modulators. Thus, we assessed the association between centiloid, each potential modulator (i.e., sTREM2, age, ApoE4), and sex on cross-sectional or longitudinal p-tau$_{181}$ levels. We detected a significant centiloid x sTREM2 x sex interaction on cross-sectional p-tau$_{181}$ (T = 2.316, *p* = 0.021; partial R$^2$ = 0.012; Fig. 3C), where higher sTREM2 levels in women were associated with higher p-tau$_{181}$ levels at a given level of Aβ. However, the analysis only reached borderline significance after FDR correction (pFDR = 0.063). Age or ApoE4 did not show interaction effects with sex on Aβ-dependent p-tau$_{181}$ levels (age: T = 0.009, *p* = 0.993; ApoE4: T = 0.362, *p* = 0.718) or p-tau$_{181}$ increases (age: T = 0.357, *p* = 0.721; ApoE4: T = 0.476, *p* = 0.634). The results are summarized in Table 3. These findings indicate that women show a trend for a different effect of sTREM2-related microglial activation on the Aβ to p-tau axis than men, which might result in higher tau burden and worsened disease progression in women.

## Discussion

In the present study, we systematically assessed modulating factors of the Aβ to p-tau axis, i.e., a potentially critical driver of tau

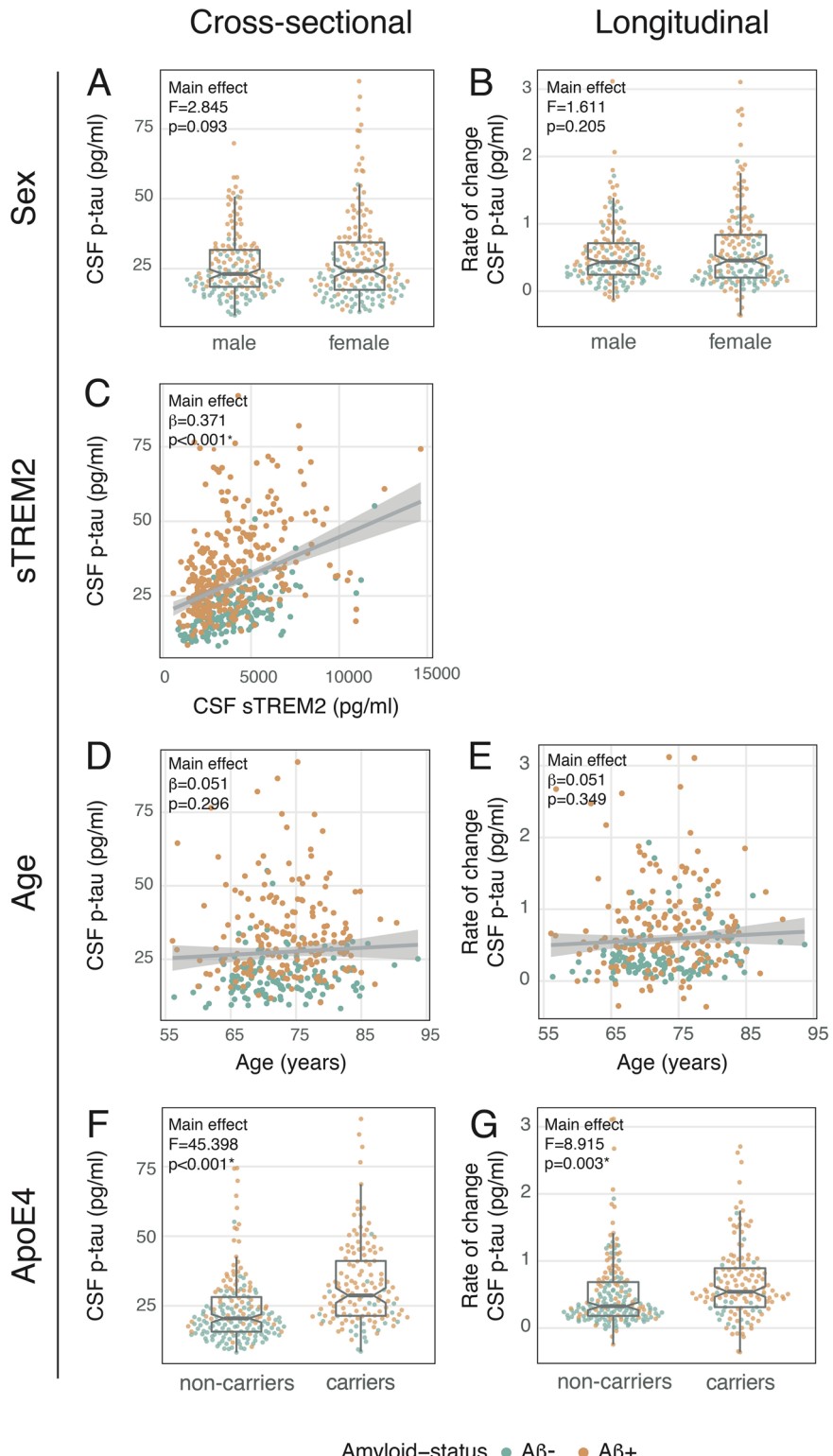

fibrillization in AD (Pichet Binette et al, 2022), to better understand heterogeneity in pathophysiological disease progression. Specifically, we assessed whether sex, microglial activation (i.e., sTREM2), age or genetic predisposition for AD (i.e., ApoE4) are associated with higher levels in p-tau$_{181}$, and specifically, modulate the Aβ to

p-tau axis. Since women are at increased risk of AD, we further tested whether effects of any potential Aβ to p-tau modulator are stronger in women than in men. First, we show that sTREM2 and ApoE4 are associated with higher p-tau$_{181}$ levels and that the effects were even more pronounced in women than in men. Second, we

◀ **Figure 1.** Main effects of potential modulators and CSF p-tau$_{181}$ across Aβ− cognitively normal plus Aβ+ ranging from cognitively normal to AD demented (longitudinal sample: N = 322; sTREM2 sample: N = 454).

Significant main effects were found between CSF sTREM2 and cross-sectional CSF p-tau$_{181}$ (C, $p < 2e{-}16$) and between ApoE4 carriership (non-carriers: $n = 185$; carriers: $n = 137$) and cross-sectional (F, $p = 7.69e{-}11$) and longitudinal (G) p-tau$_{181}$ levels. Sex (male: $n = 156$; female: $n = 166$) (A, B) and age (D, E) were not associated with higher p-tau$_{181}$ levels or faster p-tau$_{181}$ increases. F and $p$-values were derived for nominal variables (A, B, F, G) from ANCOVAs, and standardized beta-estimates (β) and $p$-values were derived for metric variables (C, D, E) from linear regressions. (A) and (B) were controlled for age, education, clinical status, and amyloid-PET (in centiloid); (C), (F), and (G) were controlled for sex, age, education, clinical status, and centiloid; (D) and (E) were controlled for sex, education, clinical status, and centiloid. The center line of each boxplot (A, B, F, G) indicates the median (50th percentile), the bounds of the box show the interquartile range (25–75th percentiles), and the whiskers extend to 1.5 times the interquartile range from the edges of the box. * = $p_{FDR} < 0.05$.

**Table 2. Main and interaction effects of potential modulators on cross-sectional and longitudinal CSF p-tau$_{181}$ levels.**

| Model | Covariates | Cross-sectional | | | Longitudinal | | |
|---|---|---|---|---|---|---|---|
| **Main effects modulators on p-tau** | | | | | | | |
| | | *ANCOVA* | | | | | |
| | | F | p | | F | p | |
| CSF p-tau$_{181}$ ~ sex | Age, education, clinical status, CL | 2.845 | 0.093 | | 1.611 | 0.205 | |
| CSF p-tau$_{181}$ ~ ApoE4 | Sex, age, education, clinical status, CL | 45.398 | <0.001* | | 8.915 | 0.003* | |
| | | *Linear regression* | | | | | |
| | | β | T | p | β | T | p |
| CSF p-tau$_{181}$ ~ sTREM2 | Sex, age, education, clinical status, CL | 0.371 | 10.098 | <0.001* | | | |
| CSF p-tau$_{181}$ ~ age | Sex, education, clinical status, CL | 0.051 | 1.047 | 0.296 | 0.051 | 0.938 | 0.349 |
| **Interactions CL x modulators on p-tau** | | β | T | p | β | T | p |
| CSF p-tau$_{181}$ ~ CL*sex | Age, education, clinical status | 0.206 | 2.746 | 0.006* | 0.182 | 2.169 | 0.031# |
| CSF p-tau$_{181}$ ~ CL*CSF sTREM2 | Sex, age, education, clinical status | 0.037 | 0.422 | 0.673 | | | |
| CSF p-tau$_{181}$ ~ CL*age | Sex, education, clinical status | −0.137 | −0.240 | 0.810 | 0.327 | 0.516 | 0.606 |
| CSF p-tau$_{181}$ ~ CL*ApoE4 | Sex, age, education, clinical status | −0.045 | −0.513 | 0.608 | −0.138 | −1.409 | 0.160 |

Main effects of sex and ApoE4 on p-tau$_{181}$ were calculated using ANCOVAs and main effects of CSF sTREM2 and age on p-tau$_{181}$ using linear regressions. Interaction effects between modulators and the association between amyloid-PET (in centiloid; CL) and p-tau$_{181}$ levels were calculated using linear regressions. * = $p_{FDR} < 0.05$; # = $p_{FDR} < 0.1$.

show that solely sex modulates Aβ-dependent p-tau$_{181}$ levels, with stronger Aβ-dependent p-tau$_{181}$ secretion in women compared to men. Finally, we observed a trend for sex differences for the association between sTREM2 and Aβ-dependent p-tau$_{181}$ levels, again, with higher p-tau$_{181}$ levels in women than in men. Together, our results underline sex-specific dynamics in AD disease pathways with more severe consequences for higher p-tau$_{181}$ levels in women. The findings are critical for patient stratifications in clinical trials, especially for drugs targeting microglial activation as a disease modifying approach.

Previously, higher levels of soluble (Hohman et al, 2018; Tsiknia et al, 2022) and aggregated tau (Buckley et al, 2019b; Shokouhi et al, 2020) were found in women compared to men. Congruently, we found that women showed stronger Aβ-related p-tau$_{181}$ secretion and trend to have faster Aβ-related p-tau$_{181}$ increases than men ($p_{FDR} = 0.092$), with sex being the only tested factor that modulated the Aβ to p-tau axis. However, future studies are needed to further investigate the effect of sex on Aβ-related longitudinal p-tau propagation. Besides a main effect of sTREM2-related microglial activation on p-tau$_{181}$ levels, we further revealed a stronger association between sTREM2 and p-tau$_{181}$ in women than in men. When testing whether this observation could be

applied to the Aβ to p-tau axis, we found a 3-way interaction of Aβ, sTREM2, and sex on p-tau$_{181}$, showing that in women, higher levels of sTREM2 were linked to a stronger Aβ-dependent p-tau$_{181}$ response. However, when applying FDR correction, the result only reached borderline significance ($p_{FDR} = 0.063$), hence, future studies are needed to confirm this finding. In our previous work investigating disease stage-dependent effects of sTREM2-related microglial activation on p-tau$_{181}$ increases, we found that in patients within earliest Aβ pathology (defined as Aβ CSF positive and amyloid-PET negative) (Palmqvist et al, 2017), sTREM2 mediated Aβ-related p-tau$_{181}$ increases (Biel et al, 2023). In addition, early Aβ pathology was associated with glucose hypermetabolism, indicating that sTREM2 follows earliest Aβ fibrillization, which might manifest in activated microglia consuming more glucose (Xiang et al, 2021). Our current observations align well with findings using TSPO-PET as a proxy of microglial activation, where TSPO-PET was associated with higher tau-PET signals in female but not in male AD patients (Biechele et al, 2024). Similarly, in post-mortem investigations, microglial activation was linked to Aβ-related tau pathology in women but not in men (Casaletto et al, 2022). Hence, microglial-induced tau accumulation might be more pronounced in women than in men.

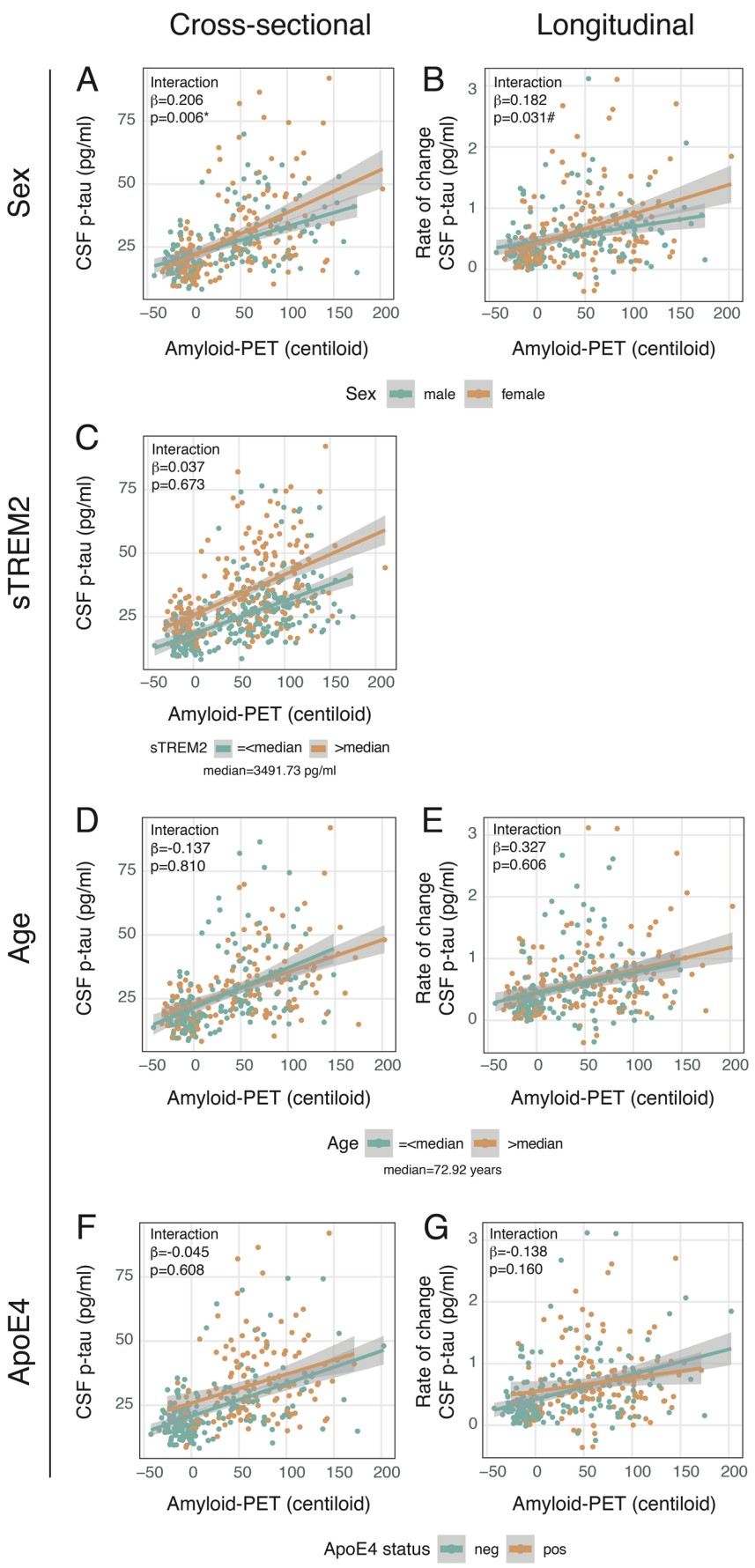

◀ **Figure 2. Interaction effects of potential modulators on the association between amyloid-PET (in centiloid) and CSF p-tau$_{181}$ across Aβ− cognitively normal plus Aβ+ ranging from cognitively normal to AD demented (longitudinal sample: N = 322; sTREM2 sample: N = 454).**

A significant interaction effect was found between centiloid and sex (male: n = 156; female: n = 166) on cross-sectional (**A**) and longitudinal (**B**) p-tau$_{181}$ levels (B only borderline significant after FDR correction). No significant interaction effects were found between centiloid and sTREM2 (**C**), age (**D, E**) or ApoE4 status (non-carriers: n = 185; carriers: n = 137) (**F, G**) on cross-sectional/longitudinal p-tau$_{181}$ levels. Standardized beta-estimates (β) and p-values were derived from linear regressions. (**A**) and (**B**) were controlled for age, education, and clinical status; (**C**), (**F**), and (**G**) were controlled for sex, age, education, and clinical status, and (**D**) and (**E**) were controlled for sex, education, and clinical status. * = $p_{FDR} < 0.05$; # = $p_{FDR} < 0.1$.

Biologically, women are predisposed to higher levels of neuroinflammatory markers than men, which has been shown using TSPO-PET across studies in healthy adults (Tuisku et al, 2019) and animal models of amyloidosis (Biechele et al, 2020). Preclinical research found that microglia have transcriptional sex differences in adult brains, suspected to be caused by sex chromosomes as well as sex hormones that might be involved in microglial functioning (Guillot-Sestier et al, 2021; Kodama and Gan, 2019; Villa et al, 2018). In addition, women are more often affected by autoimmune diseases of the central nervous system, such as multiple sclerosis (Kalincik et al, 2013; Koch-Henriksen and Sorensen, 2010), which has been further attributed to maladaptive microglial activation (Yong, 2022). Thus, sex-specific differences in microglial activation in women might result in a higher vulnerability to chronic neuroinflammation, which may cause neuronal damage (Bivona et al, 2023; Jayaraman et al, 2021). It would be critical to assess, whether the observed association between sTREM2-related microglial activation and p-tau$_{181}$ is mediated by pro-inflammatory cytokines (e.g., Interleukin 1β), which might result from prolonged microglial activation as a response to earliest Aβ fibrillization (Wang et al, 2015). Here, it should be tested, whether the threshold for a pro-inflammatory cytokine response differs between men and women. In addition, future studies should include additional markers related to neuroinflammation in the CSF (e.g., YKL-40, ICAM-1, VCAM1) (Janelidze et al, 2018; Popp et al, 2017) or tissue (e.g., TSPO-PET, FDG-PET) (Xiang et al, 2021) to address the underlying mechanism that link neuroinflammation and p-tau secretion. Moreover, future investigations could benefit from incorporating proteomic analyses to further explore more complex patterns of neuroinflammation markers, particularly those related to microglial activation. Detrimental effects of microglial activation in early disease stages of AD are in contrast to the observed positive effects microglial activation might have in later stages of AD, such as protective effects on Aβ pathology, neurodegeneration, and cognitive decline (Ewers et al, 2020; Ewers et al, 2019; Morenas-Rodriguez et al, 2022). We therefore suggest to consider subanalyses in clinical trials that are stratified by sex and disease stage, as our findings indicate that men and women show different dynamics in AD disease pathways. These might be driven among others by a different response to microglial activation and presumably earlier thresholds for Aβ-related neuroinflammation in women than in men.

Besides the association between sTREM2-related microglial activation and p-tau$_{181}$, we observed a main effect of ApoE4 status on cross-sectional p-tau$_{181}$ as well as longitudinal p-tau$_{181}$ increases, which supports our previous work showing that ApoE4 enhances tau spreading using tau-PET (Steward et al, 2023). From a mechanistic point of view, it has been shown that the ApoE4 allele interferes with the absorption of polyunsaturated fatty acids, which are vital for the cell's functioning. As a consequence of ApoE4, fewer nutrients can be absorbed, the cells become inflamed and ultimately die (Asaro et al, 2020). This inflammatory reaction might promote the secretion of soluble tau, similar to the observed effects of sTREM2-related microglial activation on p-tau (Biel et al, 2023). Indeed, previous work reported that ApoE4 facilitates microglia-related neuroinflammation and thereby might contribute to AD pathways (Kang et al, 2018; Krasemann et al, 2017; Parhizkar and Holtzman, 2022; Tai et al, 2015; Ulrich et al, 2018). Specifically, it was recently shown that ApoE4 activates microglia within brain regions that are prone to early tau propagation, and this effect was independent of Aβ (Ferrari-Souza et al, 2023). In addition, and in line with previous observations (Altmann et al, 2014; Babapour Mofrad et al, 2020; Damoiseaux et al, 2012; Hohman et al, 2018), we found that female ApoE4 risk allele carriers show higher levels of cross-sectional p-tau$_{181}$ compared to male ApoE4 risk allele carriers. Importantly, higher microglia-induced inflammatory states were previously found in female ApoE4 carriers compared to male ApoE4 carriers (Mhatre-Winters et al, 2022), suggesting similar sex-specific associations between sTREM2- and ApoE4-related neuroinflammation and p-tau$_{181}$ levels. However, with the data of the present study, causal conclusions are limited, thus, future work is needed to test the link between ApoE4-induced cell inflammation and subsequent p-tau secretion. Further, our finding of increased p-tau$_{181}$ in ApoE4 carriers may reflect, in part, their predisposition toward earlier Aβ pathology onset and thus a more advanced disease stage (Therriault et al, 2021). Given that ApoE4 carriers in our study show higher Aβ positivity and load (Figs. 1 and 2), the observed association with p-tau$_{181}$ could be influenced by their progression along the amyloid cascade. Future longitudinal studies with stage-specific controls would help clarify whether this relationship is independent of ApoE4's effects on Aβ progression. In contrast, ApoE4 status did not modulate the association between Aβ and p-tau$_{181}$, which seems surprising since ApoE4 has been extensively identified as a driver of Aβ pathology (Liu et al, 2017; Morris et al, 2010; Reiman et al, 2009). Similarly, the interaction between ApoE4 and sex on p-tau$_{181}$ was no longer present when including Aβ as interaction term. Recently, we found that Aβ mediates the association between ApoE4 and faster tau accumulation in regions that are vulnerable for early tau aggregation (Steward et al, 2023), thus, ApoE4 might only drive Aβ-related p-tau increases in early AD disease stages, while in later disease, the effects of ApoE4 on p-tau might be independent of Aβ fibrillization. Along the same lines, it has been noted that the interaction between ApoE4 and sex on p-tau levels is only persistent in early disease stages (subjective cognitive decline and MCI, but not in dementia) (Babapour Mofrad et al, 2020), consistent with another study reporting an ApoE4 x sex interaction on tau-PET only within early regions of

## A

**Female sex is associated with a stronger association between CSF sTREM2 and CSF p-tau**

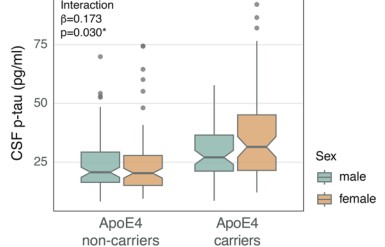

## B

**Stronger association between female ApoE4 carriers and CSF p-tau**

## C

**Trend for stronger association between CSF sTREM2 and Aβ-related CSF p-tau in women**

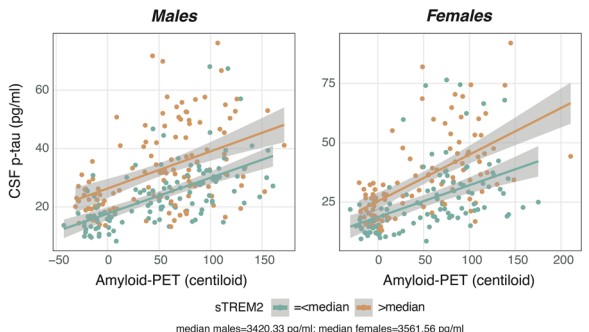

**Figure 3. Sex as modulator for the effects of CSF sTREM2 and ApoE4 carriership on CSF p-tau$_{181}$ across Aβ− cognitively normal plus Aβ+ ranging from cognitively normal to AD demented (ApoE4 sample: $N = 322$; sTREM2 sample: $N = 454$).**

Female sex compared to male sex (male: $n = 244$; female: $n = 210$) is associated with a stronger association between sTREM2 and p-tau$_{181}$ levels (**A**). Stronger association between ApoE4 carriership (non-carriers: $n = 185$; carriers: $n = 137$) and p-tau$_{181}$ in women than in men (**B**). 3-way interaction effect of centiloid, sTREM2, and sex on p-tau$_{181}$ levels with women showing a stronger association between centiloid and p-tau$_{181}$ with increasing sTREM2 levels (β = 0.312, T = 2.316, $p = 0.021$, $p_{FDR} = 0.063$) (**C**). Standardized beta-estimates (β) and $p$-values were derived from linear regressions. (**A**) and (**B**) were controlled for age, education, clinical status, and clinical status. (**C**) was controlled for age, education, and clinical status. The center line of the boxplot (**B**) indicates the median (50th percentile), the bounds of the box show the interquartile range (25–75th percentiles), and the whiskers extend to 1.5 times the interquartile range from the edges of the box. * = $p_{FDR} < 0.05$.

tau deposition (Wang et al, 2021). In the context of microglial activation and neuroinflammation, we also previously showed that sTREM2 mediates Aβ-related p-tau$_{181}$ increases only in early Aβ fibrillization (Biel et al, 2023). It would be a key next step to test whether the observed sex interactions with sTREM2 and ApoE4 on

p-tau$_{181}$ are disease stage-dependent and whether they are related to each other.

Finally, we did not observe any effects of age on cross-sectional or longitudinal p-tau$_{181}$ levels, neither a main effect nor an interaction with Aβ or sex. Older age is the major risk factor for developing AD (Guerreiro and Bras, 2015; Hou et al, 2019). However, younger age at symptom onset has been conversely found to be associated with accelerated tau accumulation (Smith et al, 2020), neurodegeneration (Moller et al, 2013), cognitive decline (van der Vlies et al, 2009) and higher rates of mortality in AD (Koedam et al, 2008). Therefore, we tested whether accelerated spread of tau in younger patients can be further explained by a stronger association between Aβ and p-tau$_{181}$, which was not confirmed by our analyses. Therefore, faster tau accumulation in younger AD patients is unlikely driven by a higher Aβ-related p-tau response and it is unclear why tau accumulates in younger patients at a faster rate and whether female patients with an early onset show a faster spreading of tau. Here, future research should further address the underlying mechanisms of tau accumulation in patients with an earlier disease onset.

A strength of this study is the inclusion of several biomarker assessments in relation to sex differences in AD, investigating modulating effects on the Aβ to p-tau axis within a large sample ranging from cognitively normal to demented. However, several limitations should be addressed when interpreting our data. First, sTREM2 data have been obtained relatively late in the ADNI dataset, hence, data with subsequent CSF p-tau$_{181}$ for assessing longitudinal relationships were limited. Since we previously show that sTREM2 mediated the association between Aβ and p-tau$_{181}$ only in earliest Aβ fibrillization, we restricted the analysis on cross-sectional data for the sTREM2 sample in order to increase the sample covering Aβ− controls to Aβ+ patients across the AD spectrum to 454 participants. We encourage future studies to assess sex differences in longitudinal associations between Aβ, sTREM2, and p-tau$_{181}$ once more data are available. Second, sTREM2 is only an indirect marker of microglial activation, thus other direct (e.g., post-mortem) or indirect (e.g., TSPO-PET) marker should be included in future studies. Third, in the present study, not sufficient tau-PET data were available to reliably test whether the stronger p-tau$_{181}$ response in women promotes faster spreading of tau aggregates. Once more data are available, this might be an important target for future investigations. Finally, the analysis should be replicated in other cohorts than ADNI and include more diverse participant groups with different ethnicities in order to increase generalizability of our findings.

In conclusion, our findings show that sex is an important modulator of Aβ-dependent p-tau$_{181}$ secretion. In particular, women show a stronger association between sTREM2-related microglial activation and p-tau$_{181}$ than men, supporting the view that neuroinflammation may play a key role in the observed sex differences in AD. In addition, we found higher p-tau$_{181}$ levels in female ApoE4 carriers compared to male ApoE4 carriers, which might be as well attributable to greater neuroinflammatory effects of ApoE4 in women. Our study provides evidence for sex-specific dynamics of AD disease pathways related to neuroinflammation and has potential implications for drug treatments targeting microglial activation, in which sex should be considered as an important modulating factor.

# Methods

### Reagents and tools table

| Reagent/Resource | Reference or Source | Identifier or Catalog Number |
|---|---|---|
| **Experimental models** | | |
| Alzheimer's disease cohort sample | Alzheimer's Disease Neuroimaging Initiative (ADNI) database (adni.loni.usc.edu) | Phase ADNI2, ADNI3, ADNIGO |
| **Recombinant DNA** | | |
| Not applicable | | |
| **Antibodies** | | |
| Biotinylated polyclonal goat anti-human TREM2 antibody | R&D Systems BAF1828 | Suarez-Calvet et al (2016) sTREM2 cerebrospinal fluid levels are a potential biomarker for microglia activity in early-stage Alzheimer's disease and associate with neuronal injury markers. EMBO Mol Med 8:466–476 |
| Monoclonal mouse anti-human TREM2 antibody | Santa Cruz Biotechnology; B-3, sc373828 | |
| SULFO-TAG-labeled anti-mouse secondary antibody | Meso Scale Discovery | |
| **Oligonucleotides and other sequence-based reagents** | | |
| Not applicable | | |
| **Chemicals, Enzymes and other reagents** | | |
| Not applicable | | |
| **Software** | | |
| R statistical software version 4.0.2 Statistical analysis: lmer, aov, lm.beta, lm Visualization: ggplot | http://www.R-project.org; R Core Team, 2021. R: A language and environment for statistical computing. R Foundation for Statistical Computing, Vienna, Austria. | |
| **Other** | | |

## Participants

To determine the effects of sex, age, and ApoE4 on cross-sectional and longitudinal p-tau$_{181}$ levels, we included 322 participants from the Alzheimer's Disease Neuroimaging Initiative (ADNI) with availability of $^{18}$F-florbetapir/$^{18}$F-florbetaben amyloid-PET, ApoE4 status, demographic information (sex, age, education), clinical status as well as longitudinal p-tau$_{181}$ data (>1 measurement). Baseline data had to be obtained within a time window of 6 months for participants to be included in this study. Participants were classified as ApoE4 risk allele carriers when at least one ε4 allele was present. For specifically assessing the effects of sTREM2, we included a larger sample of 454 ADNI participants with CSF sTREM2 in addition to available $^{18}$F-florbetapir/$^{18}$F-florbetaben amyloid-PET, demographic information (sex, age, education), and clinical status. Only cross-sectional p-tau$_{181}$ data were included for the sTREM2 sample since longitudinal p-tau$_{181}$ was not sufficiently available to specifically assess effects of sTREM2-related microglial activation on p-tau$_{181}$ change rates. Clinical status was defined by ADNI investigators as cognitively normal (CN; Mini Mental State Examination [MMSE] ≥ 24, Clinical Dementia Rating [CDR] = 0, non-depressed), mild cognitive impairment (MCI; MMSE ≥ 24, CDR = 0.5, objective memory-impairment on education-adjusted Wechsler Memory Scale II, preserved activities of daily living) or dementia (MMSE = 20–26, CDR ≥ 0.5, NINCDS/ADRDA criteria for probable AD). Aβ status (−/+) was determined using tracer-specific cut-offs for global amyloid-PET (i.e., Aβ+ = SUVR > 1.11/1.08 for $^{8}$F-florbetapir (Landau et al, 2012)/$^{18}$F-florbetaben (Royse et al, 2021)). The study is in accordance with the Declaration of Helsinki. Ethical approval was obtained by ADNI investigators, and all study participants provided written informed consent.

## CSF acquisition

The ADNI biomarker core team at the University of Pennsylvania assessed CSF p-tau$_{181}$ data using an electrochemiluminiscence immunoassays Elecsys on a fully automated Elecsys cobas e 601

**Table 3. Effects of sex for the association between potential modulators and cross-sectional/longitudinal CSF p-tau$_{181}$ and between potential modulators and Aβ-related cross-sectional/longitudinal p-tau$_{181}$ levels.**

| Model | Covariates | Cross-sectional | | | Longitudinal | | |
|---|---|---|---|---|---|---|---|
| | | β | T | p | β | T | p |
| **Interactions modulators x sex on p-tau** | | | | | | | |
| CSF p-tau$_{181}$ ~ CSF sTREM2*sex | Age, education, clinical status, CL | 0.179 | 2.177 | 0.030* | | | |
| CSF p-tau$_{181}$ ~ age*sex | Education, clinical status, CL | −0.542 | −1.005 | 0.315 | −0.442 | −0.735 | 0.463 |
| CSF p-tau$_{181}$ ~ ApoE4*sex | Age, education, clinical status, CL | 0.173 | 2.185 | 0.030* | 0.100 | 1.119 | 0.264 |
| **Interactions CL x modulators x sex on p-tau** | | | | | | | |
| CSF p-tau$_{181}$ ~ CL*sTREM2*sex | Age, education, clinical status | 0.312 | 2.316 | 0.021# | | | |
| CSF p-tau$_{181}$ ~ CL*age*sex | Education, clinical status | 0.008 | 0.009 | 0.993 | 0.352 | 0.357 | 0.721 |
| CSF p-tau$_{181}$ ~ CL*ApoE4*sex | Age, education, clinical status | 0.051 | 0.362 | 0.718 | 0.075 | 0.476 | 0.634 |

Interaction effects between potential modulators and sex on p-tau$_{181}$ as well as interaction effects between amyloid-PET (in centiloid; CL), potential modulators, and sex on p-tau$_{181}$ levels were calculated using linear regressions. * = $p_{FDR}$ < 0.05; # = $p_{FDR}$ < 0.1.

instrument and a single lot of reagents for each biomarker. sTREM2 data were assessed using a previously described ELISA approach using the MSD Platform (Ewers et al, 2019; Kleinberger et al, 2014; Suarez-Calvet et al, 2016) and are provided in the ADNIHAASSWASHULAB.csv file on the ADNI database (variable "MSDSTREM2CORRECTED"). For a detailed description of the applied methods, see https://ida.loni.usc.edu.

## MRI and PET acquisition and preprocessing

ADNI acquired 3T structural MRI of T1-weighted MPRAGE sequences using unified scanning protocols (http://adni.loni.usc.edu/methods/mri-tool/mri-analysis/). Amyloid-PET was recorded 50–70 min after $^{18}$F-florbetapir injection in $4 \times 5$ min frames or 90–110 min after $^{18}$F-florbetaben injection in $4 \times 5$ min frames. The recorded time frames were motion corrected and averaged to obtain mean images (http://adni.loni.usc.edu/methods/pet-analysis-method/pet-analysis/). Next, the Advanced Normalization Tools (ANTs (Avants et al, 2011)) high-dimensional warping algorithm was used to estimate nonlinear spatial normalization parameters based on structural skull-stripped T1-weighted images. After amyloid-PET images were co-registered to native-space T1-weighted images, the ANTs-derived normalization parameters were used to normalize the images to Montreal Neurological Institute (MNI) space. Amyloid-PET SUVRs were intensity normalized to the whole cerebellum. Global amyloid-PET SUVRs were transformed to the centiloid scale in order to harmonize between $^{18}$F-florbetapir and $^{18}$F-florbetaben Aβ tracers (Klunk et al, 2015).

## Statistical analyses

All statistical analyses were performed using R statistical software version 4.0.2 (http://www.R-project.org) (R Core Team, 2021).

Baseline characteristics between controls (i.e., CN Aβ−) and Aβ+ were compared using ANOVAs for continuous and chi-squared tests for categorial data. Except for the CSF sTREM2 analysis where only cross-sectional data were available, all analyses were performed both on cross-sectional and longitudinal CSF p-tau$_{181}$. Longitudinal p-tau$_{181}$ was determined as the annual rate of change in p-tau$_{181}$ using a linear mixed model with time from baseline as the independent variable, incorporating random intercepts and slopes to account for individual variability, with a covariance term between the random intercept and slope for each individual (Biel et al, 2023; Pichet Binette et al, 2022; Preische et al, 2019).

First, we tested the main effects of potential Aβ to p-tau modulators on cross-sectional and longitudinal p-tau$_{181}$ levels. We calculated ANCOVAs for nominal potential modulators (i.e., sex, ApoE4) and linear regressions for metric potential modulators (i.e., sTREM2, age). Potential Aβ to p-tau modulators were included as independent variables and cross-sectional or longitudinal p-tau$_{181}$ as dependent variable. The models were controlled for sex, age, education, clinical status, and amyloid-PET (i.e., centiloid), whereby sex and age were not included as covariates in the models testing sex or age as independent variable, respectively. For our main analysis, we aimed to investigate modulating effects of sex, sTREM2, age, and ApoE4 on the association between centiloid and p-tau$_{181}$. To that end, we ran separate linear regressions, including centiloid and potential modulators as independent variables and cross-sectional or longitudinal p-tau$_{181}$ as

### The paper explained

**Problem**

In Alzheimer's disease (AD), the accumulation of Aβ pathology triggers the secretion of phosphorylated tau (p-tau) from neurons which drives the aggregation and spread of fibrillar tau pathology ensuing neurodegeneration and cognitive decline. Therefore, Aβ-related increases in p-tau are a critical event in the amyloid cascade triggering the progression of AD. Thus, it is of particular importance to characterize potential modulators of Aβ-related p-tau increases which may alter downstream tau aggregation and spread. Previous studies have linked female sex, microglial activation, younger age, and the genetic risk factor ApoE4 to altered tau levels in AD. Thus, we systematically tested whether these factors modulate the association between baseline levels of PET-assessed Aβ pathology and cross-sectional or longitudinal p-tau increases in the CSF.

**Results**

We found that higher amyloid-PET was linked to elevated p-tau where only female sex compared to male sex was associated with higher Aβ-related p-tau levels (i.e., interaction between amyloid-PET and sex). In contrast, higher sTREM2 levels and ApoE4 positivity were related to constitutively elevated p-tau across the low to high Aβ spectrum (i.e., main effect), while there was no effect of sex or age. However, sex further modulated the observed main effects, with higher sTREM2-related p-tau levels in women compared to men and a stronger effect in female ApoE4 risk allele carriers on p-tau levels compared to their male counterparts.

**Impact**

Together, our results show that ApoE4 and sTREM2 are associated with higher p-tau levels, with stronger effects in women. Of factors known to modulate tau pathology in AD, sex alone modulated the Aβ to p-tau axis, where female sex is associated with a stronger Aβ-dependent p-tau secretion, potentially driven by elevated sTREM2-related microglial activation. These findings help to understand heterogenous disease progression in men and women which might be critical for clinical trials, especially for drug treatments targeting microglial activation as a disease modifying approach.

dependent variable. Covariates of the models were sex, age, education, and clinical status, whereby sex and age were not included in the models testing sex or age as independent variable, respectively. Please refer to Table 2 for a depiction of the statistical equations used within each model.

To further assess whether effects differ between men and women, we repeated the analyses for the main and interaction effects adding sex as interaction term. First, we assessed modulator (i.e., sTREM2, age, ApoE4) x sex interactions on cross-sectional or longitudinal p-tau$_{181}$, including potential modulators and sex as independent and cross-sectional or longitudinal p-tau$_{181}$ as dependent variable. The models were controlled for age, education, clinical status, and centiloid. Next, we examined 3-way interactions of centiloid, potential modulators, and sex on cross-sectional or longitudinal p-tau$_{181}$. Here, we included centiloid, potential modulators, and sex as independent variables and cross-sectional or longitudinal p-tau$_{181}$ as dependent variable. The models were controlled for age, education, and clinical status. Age was not included in the model testing age as independent variable. Please refer to Table 3 for a depiction of the statistical equations used within each analysis.

To assess effect sizes for significant results, partial $R^2$ values were calculated for linear models and partial eta-squared $(\eta^2)$ for ANCOVAs.

Each of our four analyses (i.e., main effects, CL x modulator, modulator x sex, CL x modulator x sex) was corrected for multiple comparisons for cross-sectional and longitudinal measurements separately using the false discovery rate (FDR) approach within the p.adjust R function.

## Data availability

All data used in this manuscript are publicly available from the ADNI database (adni.loni.usc.edu) upon registration and compliance with the data use agreement. The data that support the findings of this study are available upon reasonable request from the corresponding author.

The source data of this paper are collected in the following database record: biostudies:S-SCDT-10_1038-S44321-024-00190-3.

## Peer review information

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

## Acknowledgements

Data used in preparation of this article were obtained from the Alzheimer's Disease Neuroimaging Initiative (ADNI) database (adni.loni.usc.edu). As such,

the investigators within the ADNI contributed to the design and implementation of ADNI and/or provided data but did not participate in analysis or writing of this report. A complete listing of ADNI investigators can be found at: http://adni.loni.usc.edu/wp-content/uploads/howtoapply/ADNIAcknowledgementList.pdf. We thank the ADNI participants and their families who made this study possible. The study was funded by grants from the LMU (FöFoLe, 1119, awarded to DB), the Hertie foundation for clinical neurosciences (awarded to NF), and the SyNergy excellence cluster (EXC 2145/ID 390857198).

## Author contributions

**Davina Biel**: Conceptualization; Data curation; Formal analysis; Investigation; Writing—original draft; Writing—review and editing. **Marc Suárez-Calvet**: Writing—review and editing. **Anna Dewenter**: Writing—review and editing. **Anna Steward**: Writing—review and editing. **Sebastian N Roemer**: Writing—review and editing. **Amir Dehsarvi**: Writing—review and editing. **Zeyu Zhu**: Writing—review and editing. **Julia Pescoller**: Writing—review and editing. **Lukas Frontzkowski**: Writing—review and editing. **Annika Kreuzer**: Writing—review and editing. **Christian Haass**: Data curation; Writing—review and editing. **Michael Schöll**: Writing—review and editing. **Matthias Brendel**: Conceptualization; Writing—review and editing. **Nicolai Franzmeier**: Conceptualization; Data curation; Formal analysis; Investigation; Writing—original draft; Writing—review and editing.

Source data underlying figure panels in this paper may have individual authorship assigned. Where available, figure panel/source data authorship is listed in the following database record: biostudies:S-SCDT-10_1038-S44321-024-00190-3.

## Funding

## Disclosure and competing interests statement

The authors declare no competing interests.

