## [Peer Review File · EMBO Molecular Medicine]

Female sex is linked to a stronger association between sTREM2 and CSF p-tau in Alzheimer's disease

Davina Biel, Marc Suárez-Calvet, Anna Dewenter, Anna Steward, Sebastian Roemer, Amir Dehsarvi, Zeyu Zhu, Julia Pescoller, Lukas Frontzkowski, Annika Kreuzer, Christian Haass, Michael Schöll, Matthias Brendel, Nicolai Franzmeier for the Alzheimer's Disease Neuroimaging Initiative (ADNI)

Corresponding author: Davina Biel (davina.biel@med.uni-muenchen.de)

Review Timeline:

Submission Date:	2nd Aug 24
Editorial Decision:	2nd Oct 24
Revision Received:	18th Nov 24
Editorial Decision:	11th Dec 24
Revision Received:	18th Dec 24
Accepted:	18th Dec 24

Editor: Jingyi Hou

Transaction Report:

3rd Oct 2024

Dear Dr. Biel,

Thank you again for submitting your work to EMBO Molecular Medicine. We have now received feedback from all four referees who agreed to evaluate your manuscript. As you will see in the reports below, the referees find your study of potential interest but have raised several concerns that will need to be thoroughly addressed in a major revision of the manuscript.

The referees' recommendations are clear, so I won't repeat the points listed below. It's important to carefully address all the issues raised by the referees. Specifically, Referee #1 suggested that validating the results with an independent cohort would be beneficial, and we encourage you to pursue this if feasible. However, this is not a requirement for the acceptance of the manuscript.

We would welcome the submission of a revised version within three months for further consideration. As you may already know, our editorial policy allows in principle a single round of major revision, and it is therefore essential to provide responses to the referees' comments that are as complete as possible.

Please also contact us as soon as possible if similar work is published elsewhere. If other work is published, we may not be able to extend the revision period beyond three months.

I look forward to receiving your revised manuscript.

Kind regards,
Jingyi

Jingyi Hou
Editor
EMBO Molecular Medicine

We require:

- 1) A .docx formatted version of the manuscript text (including legends for main figures, EV figures and tables). Please make sure that the changes are highlighted to be clearly visible.
- 2) Individual production quality figure files as .eps, .tif, .jpg (one file per figure). For guidance, download the 'Figure Guide PDF': (<https://www.embopress.org/page/journal/17574684/authorguide#figureformat>).
- 3) A .docx formatted letter INCLUDING the reviewers' reports and your detailed point-by-point responses to their comments. As part of the EMBO Press transparent editorial process, the point-by-point response is part of the Review Process File (RPF), which will be published alongside your paper.
- 4) A complete author checklist, which you can download from our author guidelines (<https://www.embopress.org/page/journal/17574684/authorguide#submissionofrevisions>). Please insert information in the checklist that is also reflected in the manuscript. The completed author checklist will also be part of the RPF.

6) It is mandatory to include a 'Data Availability' section after the Materials and Methods. Before submitting your revision, primary datasets produced in this study need to be deposited in an appropriate public database, and the accession numbers and database listed under 'Data Availability'. Please remember to provide a reviewer password if the datasets are not yet public (see <https://www.embopress.org/page/journal/17574684/authorguide#dataavailability>).

12) Author contributions: You will be asked to provide CRediT (Contributor Role Taxonomy) terms in the submission system. These replace a narrative author contribution section in the manuscript.

13) A Conflict of Interest statement should be provided in the main text.

14) Every published paper now includes a 'Synopsis' to further enhance discoverability. Synopses are displayed on the journal webpage and are freely accessible to all readers. They include a short stand first (maximum of 300 characters, including space) as well as 2-5 one-sentences bullet points that summarizes the paper. Please write the bullet points to summarize the key NEW findings. They should be designed to be complementary to the abstract - i.e. not repeat the same text. We encourage inclusion of key acronyms and quantitative information (maximum of 30 words / bullet point). Please use the passive voice. Please attach these in a separate file or send them by email, we will incorporate them accordingly.

15) All Materials and Methods need to be described in the main text using our 'Structured Methods' format. According to this format, the Methods section includes a Reagents and Tools Table (listing key reagents, experimental models, software and relevant equipment and including their sources and relevant identifiers) followed by a Methods and Protocols section describing the methods, ideally using a step-by-step protocol format. The aim is to facilitate adoption of the methodologies across labs. Please download and fill our Reagents and Tools Table template (.docx), which you can find in our author guidelines: <https://www.embopress.org/page/journal/17574684/authorguide#structuredmethods>

16) Include a Reagents and Tools Table as part of the Methods section, which can be downloaded from our author guidelines (<https://www.embopress.org/page/journal/17574684/authorguide#structuredmethods>)

***** Reviewer's comments *****

Referee #1 (Remarks for Author):

In this study the authors assess the interaction of sex upon the correlative relationship among amyloid, APOE4, sTREM2 and p-tau181. The authors use cross sectional and longitudinal data from the ADNI cohort. Predictably sTREM2 and APOE4 were related to higher levels of p-tau181. Women exhibited higher p-tau181 levels as a function of Abeta centiloid and faster p-tau181 increases and a stronger association between sTREM2 and p-tau181 levels as well as APOE4 and p-tau181 levels. The authors also identified an interaction between sTREM2 x centiloid x sex on p-tau181 levels suggesting higher sTREM2 levels associated with higher p-tau181 levels in women as a function of centiloid. This study investigates an important topic of understanding the effect of sex on immune markers mediating amyloid to p-tau pathology in AD and has implications for stratifying study populations for improved detection of therapeutic outcomes or disease progression. As a stylistic issue, the study suffers from the discussion of data in the manuscript being out of sync with the order of data presentation in the figures. Scientifically, given the marginal statistical significance of interactions it would be appropriate to validate the ADNI observations in another study cohort.

Major critiques:

- 1.) The data for APOE4-sex interaction on p-tau181 levels did not appear to be provided.
- 2.) Many of the interactions were nominally significant after FDR corrections. It would be useful to validate these observations in a second sample cohort

Minor critiques:

- 1.) It may be worth re-organizing the figures to attempt to keep figures in the order in which they are discussed in the text.

Referee #2 (Comments on Novelty/Model System for Author):

Data from ADNI is used. This is a strength of the study. ADNI is one of the most comprehensive and well-established databases for AD research, providing high-quality, longitudinal data on a large cohort of participants.

Referee #2 (Remarks for Author):

The manuscript investigates how sex, microglial activation (measured via sTREM2), age, and ApoE4 status influence the relationship between A β and p-tau in Alzheimer's Disease (AD). Data from ADNI is used, and the study employs cross-sectional and longitudinal analyses to identify factors that modulate the A β to p-tau axis, with a particular focus on sex-specific differences. The combination of multiple potential modulators (sex, sTREM2, age, ApoE4) to examine their collective and interactive effects on the A β to p-tau pathway seems novel. Highlighting sex as a significant modulator, especially in the context of microglial activation and ApoE4 status, is important and consistent with findings that suggest a disproportionate impact of AD on women. The statistical analyses employed in the study appear robust and appropriate for the research questions posed. The use of ANCOVAs and linear regressions to assess main effects, along with interaction terms to explore sex-specific effects, is methodologically sound. Additionally, the correction for multiple comparisons using FDR improves the reliability of the findings. The authors appropriately discuss the correlational nature of the study, the indirect measurement of microglial activation, and the need for replication in more diverse cohorts. My only point is that the current title, "Sex Modulates the Associations Between Amyloid, sTREM2, and CSF p-tau181 in Alzheimer's Disease" may inadvertently suggest a causal relationship between sex and the observed associations among amyloid, sTREM2, and p-tau181 levels. Given that the study is correlational, the term "modulates" implies an active influence, which goes beyond the evidence provided by the observational data.

Marc Aurel Busche

Referee #3 (Comments on Novelty/Model System for Author):

The experiments all seem reasonable, although the methods are a little clunky. My principle issue some over interpretation, particularly of null findings.

Referee #3 (Remarks for Author):

In this paper, Biel and colleagues use data from ADNI to explore relationships between CSF p-tau 181, amyloid burden as measured by PET, neuroinflammation as measured by CSF sTREM2, and how key variables such as sex, age, and APOE e4 status modulate these relationships, with the primary objective being to better understand what variables change the relationship between amyloid and tau, which the authors call the amyloid-ptau axis. They looked both at baseline CSF values, as well as longitudinal rates of change using linear mixed effects models. Both main effects and interactions were tested. The key findings of this paper were: (1) APOE carriers had an increased CSF p-tau levels at baseline as well of rates of change over time; (2) there was an increased slope in the relationship between cross-sectional and longitudinal p-tau 181 and amyloid in females compared to males, and (3) there was a positive association between sTREM2 levels and baseline p-tau 181 (4) the slope of this association was higher in females than males and (5) in female individuals, increased sTREM resulted in an increase in slope between amyloid and ptau, though this three-way interaction term did not reach the pre-specified significance threshold once correcting for multiple comparisons. These results seem to indicate a differential role of sex and neuroinflammation, as measured by sTREM2, in terms of how p-tau increases, primarily in individuals with evidence of amyloid pathology.

GENERAL COMMENTS

1. Methods "adjusting for random slope and intercept" - could the authors confirm that the random effects structure of this linear mixed effects also includes a term for covariance between the slope and intercept?
2. Methods - the wording describing the models could be explained more clearly. I found the way each model is represented in Tables 2 and 3 quite helpful in understanding what each model represents, and I would recommend the authors either cross-reference that table in the Methods to help the reader better understand the models or create some kind of figure to make it easier for the reader to understand the models that are being tested.
3. There are many instances in the results section, where interpretation of the results are couched in accepting the null when the p-value is over the pre-specified threshold. For example "The results indicate that sex, but not sTREM2, age, or ApoE4 modulate the A β to p-tau axis, suggesting a stronger association between A β -dependent p-tau181 secretion and eventually faster tau aggregation in women compared to men." ,The fact that a p-value is above 0.05 does not mean that the authors can state that sTREM2, age or APOE do not modulate the A β to ptau Axis, merely that they don't have any evidence that it does, and even if it were the magnitude of the effect is likely negligible. So these sorts of wordings need to be a bit more precise. Other examples "but no faster p-tau181 increases", "Sex did not have an effect on the association", etc. Please review the results and adjust accordingly.
4. There also is an implied comparison when the authors state that one effect is solely related but not another. Perhaps if one wanted to highlight the differences in the association, instead of p-values, the authors could compare the effect sizes of these associations, particularly in the cases where the same data set was used, or in joint models if these were performed.
5. Discussion- There is a fair amount of evidence that APOE results in an earlier age of onset of amyloid pathology. How can the authors be sure that the increase in ptau in APOE carriers observed in Figure 1(e) is a direct relationship and not just down to the fact that APOE carriers tend to have earlier onset and on average are likely to be further down the disease cascade than the non-carriers? I don't see any summary of these variables broken down by APOE status, but I see in Figure 1 far more amyloid positive APOE carriers than non-carriers and Figure 2 suggests that the carriers have a higher amyloid load, so more likely to be further down the disease cascade. Could that be possibility be included in the discussion?
6. Results, Figure 3(b) - I'm not sure if stronger association is what is implied here. I can see that the slope of the association is increased in females compared to males, but in terms of strength of association, the amount of variability between lines seems

similar visually. Are the authors referring to the fact that increased levels of sTREM tend to result in higher levels of ptau in females compared to males or that more of the variability between these two variables is explained in females compared to males?

MINOR COMMENT

1. Introduction "Although the driving mechanisms behind these findings are not fully clear, differences in sex hormones(21) and inflammatory processes(21)" - I think the second reference is meant to be 22?
2. Table 1 - Consider using fewer significant digits for some of the reports.
3. Table 1 - Given that by definition these two groups are separated by amyloid status (and to a lesser extent cognition/clinical severity), is there really any reason to perform a statistical test comparing amyloid between the two groups?

Referee #4 (Comments on Novelty/Model System for Author):

The paper is straightforward and clear, but to me falls a bit short on the novelty and some of the key results are more trend level, which slightly decreases their impact.

I tried to make a few suggestions to increase the impact/novelty of the paper.

Referee #4 (Remarks for Author):

Authors investigated the moderating effect of sex, APOE4 and sTREM2 on p-tau181 and Ab associations in ADNI. The results are straightforward and clearly presented.

The authors mention in the discussion how additional markers of neuroinflammation and markers related to microglia should be investigated. I wonder if some markers of interest could be taken from the SomaLogic proteomics data available in ADNI. I can imagine that there would be some overlap with the participants with CSF p-tau181 and sTREM2. If that is the case, I think it would improve the novelty of the paper.

I also wondered if authors investigated the interaction between sTREM2 and APOE4 on p-tau, i.e. similar associations as in Figure 3, but with APOE4 instead of sex. Could it help further understand if the associations are both influenced by sex and APOE4?

Regarding the two samples looking at sex and APOE4 vs sTREM2, is it the same set of participants but with additional people in the sTREM2 sample or very different samples? If it is the latter, do you have the same associations with sTREM2 if you restrict it to the first set of participants?

I think it would be useful to better grasp the results if some statistics could be added to the Abstract.

We thank the editor and the reviewer's for their encouraging feedback on our manuscript. Below please find our replies including a description of all changes that were made to the manuscript according to the comments raised by the reviewer's. We hope that we have sufficiently addressed all concerns and believe that the manuscript has been improved and strengthened during the review process.

Referee #1 (Remarks for Author):

In this study the authors assess the interaction of sex upon the correlative relationship among amyloid, APOE4, sTREM2 and p-tau181. The authors use cross sectional and longitudinal data from the ADNI cohort. Predictably sTREM2 and APOE4 were related to higher levels of p-tau181. Women exhibited higher p-tau181 levels as a function of Abeta centiloid and faster p-tau181 increases and a stronger association between sTREM2 and p-tau181 levels as well as APOE4 and p-tau181 levels. The authors also identified an interaction between sTREM2 x centiloid x sex on p-tau181 levels suggesting higher sTREM2 levels associated with higher p-tau181 levels in women as a function of centiloid. This study investigates an important topic of understanding the effect of sex on immune markers mediating amyloid to p-tau pathology in AD and has implications for stratifying study populations for improved detection of therapeutic outcomes or disease progression. As a stylistic issue, the study suffers from the discussion of data in the manuscript being out of sync with the order of data presentation in the figures. Scientifically, given the marginal statistical significance of interactions it would be appropriate to validate the ADNI observations in another study cohort.

Major critiques:

1.) The data for APOE4-sex interaction on p-tau181 levels did not appear to be provided.

Reply: We initially decided not to include the figure for the ApoE4*sex interaction in the manuscript, since this result has been already shown in previous work (e.g. Hohman et al., 2018, JAMA Neurol.). However, for clarity, we added the respective plot along to the sTREM2*sex interaction on p-tau to figure 3, panel B.

Fig. 3:

Figure 3. Sex as modulator for the effects of CSF sTREM2 and ApoE4 carriership on CSF p-tau₁₈₁ across A β - cognitively normal plus A β + ranging from cognitively normal to AD demented (ApoE4 sample: N=322; sTREM2 sample: N=454). Female sex compared to male sex is associated with a stronger association between sTREM2 and p-tau₁₈₁ levels (**A**). Stronger association between ApoE4 carriership and p-tau₁₈₁ in women than in men (**B**). 3-way interaction effect of centiloid, sTREM2, and sex on p-tau₁₈₁ levels with women showing a stronger association between centiloid and p-tau₁₈₁ with increasing sTREM2 levels ($\beta=0.312$, $T=2.316$, $p=0.021$, $p_{FDR}=0.063$) (**C**). Standardized beta-estimates (β) and p-values were derived from linear regressions. A+B were controlled for age, education, clinical status, and centiloid. C was controlled for age, education, and clinical status. * =

$p_{FDR} < 0.05$.

2.) Many of the interactions were nominally significant after FDR corrections. It would be useful to validate these observations in a second sample cohort

Reply: It is correct that the interactions between CL*sex on longitudinal p-tau ($p=0.031$, $p_{FDR}=0.092$) as well as CL*sex*sTREM2 on p-tau ($p=0.021$, $p_{FDR}=0.063$) only reached trend level significance when applying FDR correction. We have already taken these findings into account in our wording within the results section (e.g. *“These findings indicate that women show a trend for a different effect of sTREM2-related microglial activation on the A β to p-tau axis than men”*). And pointed out to these limitations within the discussion *“(...) we found a 3-way interaction of A β , sTREM2, and sex on p-tau₁₈₁, showing that in women, higher levels of sTREM2 were linked to a stronger A β -dependent p-tau₁₈₁ response. However, when applying FDR correction, the result only reached borderline significance ($p_{FDR}=0.063$), hence, future studies are needed to confirm this finding.”*

We added an additional note after reporting the trend level significance between CL*sex on longitudinal p-tau.

P. 11: *“However, future studies are needed to further investigate the effect of sex on A β -related longitudinal p-tau propagation.”*

Further, we agree with the reviewer that replicating our findings in a different cohort would be beneficial and increase the generalizability of our findings. Unfortunately, cohorts including CSF sTREM2 assessments are very limited and rarely shared, and to date, data including sTREM2 assessments together with amyloid and p-tau are unique for the publicly available ADNI cohort. To our knowledge, no other cohort provides these data openly in addition to our other assessed markers. In the limitations section we therefore already emphasized that repeating the study in a different cohort would be desirable. However, we currently do not have access to such a dataset.

P. 14: *“We encourage future studies to assess sex-differences in longitudinal associations between A β , sTREM2, and p-tau₁₈₁ once more data are available.”* And *“Finally, the analysis should be replicated in other cohorts than ADNI and include more diverse participant groups with different ethnicities in order to increase generalizability of our findings.”*

Minor critiques:

1.) It may be worth re-organizing the figures to attempt to keep figures in the order in which they are discussed in the text.

Reply: We agree with the reviewer and re-organized the figures according to the results section (order: Sex, sTREM2, age, ApoE4).

Fig. 1:

Figure 1. Main effects of potential modulators and CSF p-tau₁₈₁ across Aβ⁻ cognitively normal plus Aβ⁺ ranging from cognitively normal to AD demented (longitudinal sample: N=322; sTREM2 sample: N=454). Significant main effects were found between CSF sTREM2 and cross-sectional CSF p-tau₁₈₁ (C) and between ApoE4 carriership and cross-sectional (F) and longitudinal (G) p-tau₁₈₁ levels. Sex (A+B) and age (D+E) were not associated with higher p-tau₁₈₁ levels or faster p-tau₁₈₁ increases. F and p-values were derived for nominal variables (A+B; F+G) from

ANCOVAs, and standardized beta-estimates (β) and p-values were derived for metric variables (C+D+E) from linear regressions. A+B were controlled for age, education, clinical status, and amyloid-PET (in centiloid); C+F+G were controlled for sex, age, education, clinical status, and centiloid; D+E were controlled for sex, education, clinical status, and centiloid. * = $p_{FDR} < 0.05$.

Fig. 2:

Figure 2. Interaction effects of potential modulators on the association between amyloid-PET (in centiloid) and CSF p-tau₁₈₁ across Aβ⁻ cognitively normal plus Aβ⁺ ranging from cognitively normal to AD demented (longitudinal sample: N=322; sTREM2 sample: N=454). A significant interaction effect was found between centiloid and sex on cross-sectional **(A)** and longitudinal **(B)** p-tau₁₈₁ levels (B only borderline significant after FDR correction). No significant interaction effects were found between centiloid and sTREM2 **(C)**, age **(D+E)** or ApoE4 status **(F+G)** on cross-sectional/longitudinal p-tau₁₈₁ levels. Standardized beta-estimates (β) and p-values were derived from linear regressions. A+B were controlled for age, education, and clinical status; C+F+G were controlled for sex, age, education, and clinical status, and D+E were controlled for sex, education, and clinical status. * = $p_{\text{FDR}} < 0.05$; # = $p_{\text{FDR}} < 0.1$.

Referee #2 (Comments on Novelty/Model System for Author):

Data from ADNI is used. This is a strength of the study. ADNI is one of the most comprehensive and well-established databases for AD research, providing high-quality, longitudinal data on a large cohort of participants.

Referee #2 (Remarks for Author):

The manuscript investigates how sex, microglial activation (measured via sTREM2), age, and ApoE4 status influence the relationship between A β and p-tau in Alzheimer's Disease (AD). Data from ADNI is used, and the study employs cross-sectional and longitudinal analyses to identify factors that modulate the A β to p-tau axis, with a particular focus on sex-specific differences. The combination of multiple potential modulators (sex, sTREM2, age, ApoE4) to examine their collective and interactive effects on the A β to p-tau pathway seems novel. Highlighting sex as a significant modulator, especially in the context of microglial activation and ApoE4 status, is important and consistent with findings that suggest a disproportionate impact of AD on women. The statistical analyses employed in the study appear robust and appropriate for the research questions posed. The use of ANCOVAs and linear regressions to assess main effects, along with interaction terms to explore sex-specific effects, is methodologically sound. Additionally, the correction for multiple comparisons using FDR improves the reliability of the findings. The authors appropriately discuss the correlational nature of the study, the indirect measurement of microglial activation, and the need for replication in more diverse cohorts. My only point is that the current title, "Sex Modulates the Associations Between Amyloid, sTREM2, and CSF p-tau181 in Alzheimer's Disease" may inadvertently suggest a causal relationship between sex and the observed associations among amyloid, sTREM2, and p-tau181 levels. Given that the study is correlational, the term "modulates" implies an active influence, which goes beyond the evidence provided by the observational data.

Marc Aurel Busche

Reply: We thank the reviewer for the overall positive and encouraging feedback and followed his suggestion in modifying the title. It now reads: *"Female sex is linked to a stronger association between sTREM2 and CSF p-tau in Alzheimer`s disease"*.

Referee #3 (Comments on Novelty/Model System for Author):

The experiments all seem reasonable, although the methods are a little clunky. My principle issue some over interpretation, particularly of null findings.

Referee #3 (Remarks for Author):

In this paper, Biel and colleagues use data from ADNI to explore relationships between CSF p-tau 181, amyloid burden as measured by PET, neuroinflammation as measured by CSF sTREM2, and how key variables such as sex, age, and APOE e4 status modulate these relationships, with the primary objective being to better understand what variables change the relationship between amyloid and tau, which the authors call the amyloid-ptau axis. They looked both at baseline CSF values, as well as longitudinal rates of change using linear mixed effects models. Both main effects and interactions were tested. The key findings of this paper were: (1) APOE carriers had an increased CSF p-tau levels at baseline as well of rates of change over time; (2) there was an increased slope in the relationship between cross-sectional and longitudinal p-tau 181 and amyloid in females compared to males, and (3) there was a positive association between sTREM2 levels and baseline p-tau 181 (4) the slope of this association was higher in females than males and (5) in female individuals, increased sTREM resulted in an increase in slope between amyloid and ptau, though this three-way interaction term did not reach the pre-specified significance threshold once correcting for multiple comparisons. These results seem to indicate a differential role of sex and neuroinflammation, as measured by sTREM2, in terms of how p-tau increases, primarily in individuals with evidence of amyloid pathology.

GENERAL COMMENTS

1. Methods "adjusting for random slope and intercept" - could the authors confirm that the random effects structure of this linear mixed effects also includes a term for covariance between the slope and intercept?

Reply: Yes, the equation we applied includes a term for the covariance between the slope and intercept in the random-effects structure by default. We decided to include a more precise wording in the respective section.

P. 6: *„Longitudinal p-tau₁₈₁ was determined as the annual rate of change in p-tau₁₈₁ using a linear mixed model with time from baseline as the independent variable, incorporating random intercepts and slopes to account for individual variability, with a covariance term between the random intercept and slope for each individual.”*

2. Methods - the wording describing the models could be explained more clearly. I found the way each model is represented in Tables 2 and 3 quite helpful in understanding what each model represents, and I would recommend the authors either cross-reference that table in the Methods to help the reader better understand the models or create some kind of figure to make it easier for the reader to understand the models that are being tested.

Reply: We thank the reviewer for this feedback. We added a cross-reference to the descriptions in Tables 2 and 3 to the respective methods sections.

P. 7: *“Please refer to Table 2 for a depiction of the statistical equations used within each model.”* and *“Please refer to Table 3 for a depiction of the statistical equations used within each analysis.”*

3. There are many instances in the results section, where interpretation of the results are couched in accepting the null when the p-value is over the pre-specified threshold. For example "The results indicate that sex, but not sTREM2, age, or ApoE4 modulate the A β to p-tau axis, suggesting a stronger association between A β -dependent p-tau₁₈₁ secretion and eventually faster tau aggregation in women compared to men." ,The fact that a p-value is above 0.05 does not mean that the authors can state that sTREM2, age or APOE do not modulate the A β to ptau Axis, merely that they don't have any evidence that it does, and even if it were the magnitude of the effect is likely negligible. So these sorts of wordings need to be a bit more precise. Other examples "but no faster p-tau₁₈₁ increases", "Sex did not have an effect on the association", etc. Please review the results and adjust accordingly.

Reply: We rephrased respective sections in the results section.

P. 9: *“The results indicate that sex is associated with modulation of the A β to p-tau axis, suggesting a stronger association between A β -dependent p-tau₁₈₁ secretion and eventually faster tau aggregation in women compared to men. In contrast, our analysis did not provide statistical evidence that sTREM2, age, or ApoE4 significantly modulate the A β to p-tau axis.”*

P. 10: *“There was no significant effect of sex on the association between age and p-tau₁₈₁ levels (T=-1.005, p=0.315) or p-tau₁₈₁ increases (T=-0.735, p=0.463).”*

In regard to the second example of the reviewer which reads as following *“We detected a significant interaction between ApoE4 carriership and sex on p-tau₁₈₁, (...) but no faster p-tau₁₈₁ increases (T=1.119, p=0.264).”* we think that

we have chosen the correct wording, as we did not detect a significant interaction (i.e. $p < 0.05$) between ApoE4, sex and p-tau increases.

4. There also is an implied comparison when the authors state that one effect is solely related but not another. Perhaps if one wanted to highlight the differences in the association, instead of p-values, the authors could compare the effect sizes of these associations, particularly in the cases where the same data set was used, or in joint models if these were performed.

Reply: We agree with the Reviewer that comparing effect sizes do allow a more meaningful interpretation than just p-values alone as they offer an important perspective on the magnitude of the observed relationships. However, we believe that p-values provide essential context in assessing statistical significance. As a result, we have chosen to emphasize p-values to determine the significance of our findings and then supplement significant results with effect size measures to provide a clearer picture of the practical impact. We also note that the t-values in our analyses can serve as additional indicators of effect size, providing further insight into the strength of relationships observed.

P. 7: *“To assess effect sizes for significant results, partial R^2 values were calculated for linear models and partial eta-squared (η^2) for ANCOVAs.”*

P. 8: *“We found that higher sTREM2 was related to higher p-tau₁₈₁ levels ($T=10.098$, $p < 0.001$, $p_{FDR} < 0.001$, partial $R^2=0.186$; **Fig.1C**), and that ApoE4 was related to higher p-tau₁₈₁ levels ($F=45.398$, $p < 0.001$, $p_{FDR} < 0.001$, $\eta^2=0.13$; **Fig.1F**) and faster p-tau₁₈₁ increases ($F=8.915$, $p=0.003$, $p_{FDR}=0.009$, $\eta^2=0.03$; **Fig.1G**).”*

P. 9: *“(…) where women compared to men showed higher A β -related p-tau₁₈₁ levels ($T=2.746$, $p=0.006$, $p_{FDR}=0.026$; partial $R^2=0.023$; **Fig.2A**) and faster p-tau₁₈₁ increases ($T=2.169$, $p=0.031$; partial $R^2=0.015$; **Fig.2B**) (…)”*

P. 9: *“(…) with women showing a stronger association between sTREM2 and p-tau₁₈₁ levels compared to men ($T=2.177$, $p=0.030$, $p_{FDR}=0.045$; partial $R^2=0.011$; **Fig.3A**).”*

P. 10: *“(…) where female ApoE4 carriers compared to male ApoE4 carriers showed higher levels of p-tau₁₈₁ ($T=2.185$, $p=0.030$, $p_{FDR}=0.045$; partial $R^2=0.015$; **Fig.3B**) (…)”*

5. Discussion- There is a fair amount of evidence that APOE results in an earlier age of onset of amyloid pathology. How can the authors be sure that the increase in ptau in APOE carriers observed in Figure 1(e) is a direct

relationship and not just down to the fact that APOE carriers tend to have earlier onset and on average are likely to be further down the disease cascade than the non-carriers? I don't see any summary of these variables broken down by APOE status, but I see in Figure 1 far more amyloid positive APOE carriers than non-carriers and Figure 2 suggests that the carriers have a higher amyloid load, so more likely to be further down the disease cascade. Could that be possibility be included in the discussion?

Reply: The reviewer raised an important point. We would like to refer to the scatterplots in Fig. 2 F+G, which show the association between CL*ApoE4 on p-tau where no interaction effect could be revealed. We have already discussed this observation within the discussion: *“In contrast, ApoE4 status did not modulate the association between A β and p-tau₁₈₁, which seems surprising since ApoE4 has been extensively identified as a driver of A β pathology.⁸⁷⁻⁸⁹ Similarly, the interaction between ApoE4 and sex on p-tau₁₈₁ was no longer present when including A β as interaction term. Recently, we found that A β mediates the association between ApoE4 and faster tau accumulation in regions that are vulnerable for early tau aggregation,⁷⁹ thus, ApoE4 might only drive A β -related p-tau increases in early AD disease stages, while in later disease, the effects of ApoE4 on p-tau might be independent of A β fibrillization.”*

In addition, we included following paragraph in the discussion.

P. 13: *“Further, our finding of increased p-tau₁₈₁ in ApoE4 carriers may reflect, in part, their predisposition toward earlier A β pathology onset and thus a more advanced disease stage.⁸⁶ Given that ApoE4 carriers in our study show higher A β positivity and load (Fig.1 and 2), the observed association with p-tau₁₈₁ could be influenced by their progression along the amyloid cascade. Future longitudinal studies with stage-specific controls would help clarify whether this relationship is independent of ApoE4's effects on A β progression.”*

6. Results, Figure 3(b) - I'm not sure if stronger association is what is implied here. I can see that the slope of the association is increased in females compared to males, but in terms of strength of association, the amount of variability between lines seems similar visually. Are the authors referring to the fact that increased levels of sTREM tend to result in higher levels of ptau in females compared to males or that more of the variability between these two variables is explained in females compared to males?

Reply: We appreciate the reviewer's observation. Our intention was to highlight that increased sTREM2 levels appear to correspond to higher levels

of p-tau in females compared to males, as suggested by the steeper slope in females. However, we recognize the importance of distinguishing between the magnitude of the association (i.e., slope) and the strength of the association (i.e., proportion of variance explained). To examine whether the strength of the association between sTREM2 and p-tau differs between sexes, we conducted separate linear regression analyses for males and females and compared the R^2 values, which indicate that the strength of the association is stronger in women than in men (women R^2 : 0.253 vs men R^2 : 0.132).

P. 9: *“Post-hoc analyses calculating separate linear regression analyses for men and women further revealed higher partial R^2 values in women ($T=8.286$, $p<0.001$, partial $R^2=0.253$) vs. men ($T=5.978$, $p<0.001$, partial $R^2=0.132$), supporting a stronger association between sTREM2 and p-tau₁₈₁ in women than in men.”*

MINOR COMMENT

1. Introduction "Although the driving mechanisms behind these findings are not fully clear, differences in sex hormones(21) and inflammatory processes(21)" - I think the second reference is meant to be 22?

Reply: We thank the reviewer for their attention and have corrected the citation.

2. Table 1 - Consider using fewer significant digits for some of the reports.

Reply: We agree with the reviewers suggestion and removed the β values from the results sections, since they are already presented in the tables of the manuscript.

3. Table 1 - Given that by definition these two groups are separated by amyloid status (and to a lesser extent cognition/clinical severity), is there really any reason to perform a statistical test comparing amyloid between the two groups?

Reply: We agree with the reviewer that the statistical difference between both amyloid groups is expected. However, for uniformity reasons we decided to stick to the p-value for the centiloid comparison between groups within the table.

Referee #4 (Comments on Novelty/Model System for Author):

The paper is straightforward and clear, but to me falls a bit short on the novelty and some of the key results are more trend level, which slightly decreases their impact. I tried to make a few suggestions to increase the impact/novelty of the paper.

Referee #4 (Remarks for Author):

Authors investigated the moderating effect of sex, APOE4 and sTREM2 on p-tau181 and Ab associations in ADNI. The results are straightforward and clearly presented.

The authors mention in the discussion how additional markers of neuroinflammation and markers related to microglia should be investigated. I wonder if some markers of interest could be taken from the SomaLogic proteomics data available in ADNI. I can imagine that there would be some overlap with the participants with CSF p-tau181 and sTREM2. If that is the case, I think it would improve the novelty of the paper.

Reply: We thank the reviewer for the insightful suggestion to investigate proteomic markers and we agree that exploring additional markers of neuroinflammation, including microglia-related markers, could offer valuable insights. While we recognize that there may indeed be an overlap with participants who have CSF p-tau₁₈₁ and sTREM2 data, integrating a comprehensive proteomics analysis is beyond the current scope of our study. Our primary focus was on factors that have been previously linked to increased tau burden (e.g. CSF sTREM2) and a detailed exploration of proteomic markers would require further methodological considerations. However, we agree that this approach holds great potential for future research and would be an exciting direction for follow-up studies. We appreciate the reviewer's suggestion and will consider it for subsequent work. We included a respective section in the discussion.

P. 12: *"In addition, future investigations could benefit from incorporating proteomic analyses to further explore more complex patterns of neuroinflammation markers, particularly those related to microglial activation."*

I also wondered if authors investigated the interaction between sTREM2 and APOE4 on p-tau, i.e. similar associations as in Figure 3, but with APOE4 instead of sex. Could it help further understand if the associations are both influenced by sex and APOE4?

Reply: Within the current project, we did not investigate the sTREM2*ApoE4 interaction on p-tau increases. When assessing the relationship within this review process, no interaction was revealed ($T=-0.410$, $p=0.682$). However, we agree with the reviewer that the association between microglial activation, ApoE4 and sex on tau pathology is a promising field of future research, since ApoE4 has been shown to activate microglia in brain regions that are prone to early tau propagation (e.g. Ferrari-Souza, Sci Adv, 2023). We acknowledged this interesting research line already in the discussion section: *“As a consequence of ApoE4, fewer nutrients can be absorbed, the cells become inflamed and ultimately die.⁸⁰ This inflammatory reaction might promote the secretion of soluble tau, similar to the observed effects of sTREM2-related microglial activation on p-tau.³⁵ Indeed, previous work reported that ApoE4 facilitates microglia-related neuroinflammation and thereby might contribute to AD pathways.^{29,81-84}”* and *“(…) we found that female ApoE4 risk allele carriers show higher levels of cross-sectional p-tau₁₈₁ compared to male ApoE4 risk allele carriers. Importantly, higher microglia-induced inflammatory states were previously found in female ApoE4 carriers compared to male ApoE4 carriers,⁸⁵ suggesting similar sex-specific associations between sTREM2- and ApoE4-related neuroinflammation and p-tau₁₈₁ levels. However, with the data of the present study, causal conclusions are limited, thus, future work is needed to test the link between ApoE4-induced cell inflammation and subsequent p-tau secretion.”*

In addition, we added respective sentence:

P13: *“Specifically, it was recently shown that ApoE4 activates microglia within brain regions that are prone to early tau propagation, and this effect was independent of A β .⁸⁵”*

Regarding the two samples looking at sex and APOE4 vs sTREM2, is it the same set of participants but with additional people in the sTREM2 sample or very different samples? If it is the latter, do you have the same associations with sTREM2 if you restrict it to the first set of participants?

Reply: The samples are partly overlapping. The cross-sectional sTREM2 sample consists of 225 participants from the longitudinal sample (N=322) plus 229 new participants. When repeating the analysis with the 225 overlapping participants, the association between sTREM2 and p-tau₁₈₁ remains significant ($T=7.294$, $p<0.001$). When assessing an sTREM2*sex interaction on p-tau₁₈₁, the association didn't reach significance ($T=1.587$, $p=0.114$). Importantly, the direction and magnitude of the association are consistent with those observed in the larger sTREM2 sample, suggesting that the smaller sample size (N=225 vs N=454) limited the statistical power.

Overlapping sample (N=225)

Original sTREM2 sample (N=454)

Female sex is associated with a stronger association between CSF sTREM2 and CSF p-tau

I think it would be useful to better grasp the results if some statistics could be added to the Abstract.

Reply: We added a respective sentence to the abstract: “Running ANCOVAs for nominal and linear regressions for metric variables, we found (...)” Due to the restriction of 175 words, some other parts in the abstract needed to be shortened.

11th Dec 2024

Dear Dr. Biel,

Thank you for submitting your revised manuscript to EMBO Molecular Medicine. We have now received the enclosed report from the three referees who re-assessed your work. As you will see, the referees are now supportive, and I am pleased to inform you that we will be able to accept your manuscript pending the following amendments:

1. Since the final article includes only three figures, we will change the manuscript type from "Research article" to "Report". According to our editorial guideline regarding the Report article type (<https://www.embopress.org/page/journal/17574684/authorguide#reportsarticleguide>), please combine the Results and Discussions sections.
2. Please remove figures from the manuscript file, but keep the legends.
3. Funding information needs to be part of "Acknowledgements". Please also move the footnote "†Data used in preparation of this article were obtained from the Alzheimer's Disease Neuroimaging Initiative (ADNI) database (adni.loni.usc.edu). As such, the investigators within the ADNI contributed to the design and implementation of ADNI and/or provided data but did not participate in analysis or writing of this report. A complete listing of ADNI investigators can be found at: <http://adni.loni.usc.edu/wp-content/uploads/howtoapply/ADNIAcknowledgementList.pdf>" to the Acknowledgements section.
4. The references need to be formatted according to the EMBO Molecular Medicine reference style.
 - Please list up to 10 co-authors of a paper before adding et al. in the reference list.
 - Citations should be listed in alphabetical order.
 - DOIs should only be used for preprints and datasets that have not been published yet.
5. "Competing interest" should be renamed to "Disclosure and competing interests statement".
6. Synopsis image: please provide the figure as a PNG file 550 px wide x 300-600 px high. The current image is too large.
7. I have slightly shortened the synopsis text(see attached). Please let me know if it is fine as is or if you would like to introduce further modifications.
8. Please move "the paper explained" to the manuscript file.
9. "Materials and methods" section should be renamed to "Methods" and placed after Discussion. The Methods section in the published papers includes a Reagents and Tools Table (listing key reagents, experimental models, software and relevant equipment and including their sources and relevant identifiers).

Please download and fill our Reagents and Tools Table template (.docx), which you can find in our author guidelines: <https://www.embopress.org/page/journal/17574684/authorguide#structuredmethods>

10. Please address the following issues regarding figure legends:
 - Please note that the exact p values are not provided in the legend of figure 1e.
 - Please note that the box plots need to be defined in terms of minima, maxima, centre, bounds of box and whiskers, and percentile in the legends of figures 1a-b, e-f; 3b.
 - Please note that information related to n is missing in the legends of figures 1a-b, e-f; 3b.

I look forward to reading a new revised version of your manuscript as soon as possible.

Kind regards,
Jingyi

Jingyi Hou
Editor
EMBO Molecular Medicine

*** Instructions to submit your revised manuscript ***

***** Reviewer's comments *****

Referee #1 (Comments on Novelty/Model System for Author):

This is a very well done study.

Referee #1 (Remarks for Author):

The authors have adequately addressed earlier comments and I recommend publication.

Referee #3 (Comments on Novelty/Model System for Author):

No substantial changes have occurred since the last iteration around the statistical modelling.

Referee #3 (Remarks for Author):

The authors have clearly reviewed the comments from reviewers carefully and provided thoughtful responses. The changes made to the manuscript have included the overall quality. I have no further recommendations.

Referee #4 (Remarks for Author):

Thank you for the revision! I think the paper has been improved and clarified.

The authors addressed the remaining editorial issues.

18th Dec 2024

Dear Dr. Biel,

We are pleased to inform you that your manuscript is accepted for publication and is now being sent to our publisher to be included in the next available issue of EMBO Molecular Medicine.

Yours sincerely,
Jingyi

Jingyi Hou
Editor
EMBO Molecular Medicine
